Overexpression of ORAOV1 and its association with immunotherapy resistance in hepatocellular carcinoma

Huang Yuzhen 1 2
Yang Ni 1 2
Wen Su 1 2
Fang Ziwei 1 2
Zhang Yucong 1 2
Qian Zonghao 1 2
Huang Yi 1 2
Yin Tiejun 1 2
Zhang Cuntai ctzhang0425@163.com 1 2
Zhang Le le_zhang@foxmail.com 1 2
1 Department of Geriatrics, Tongji Hospital, Tongji Medical College, Huazhong University of Science and Technology , Wuhan , China
2 Key Laboratory of Vascular Aging, Ministry of Education, Tongji Hospital, Tongji Medical College, Huazhong University of Science and Technology , Wuhan , China
Uversky Vladimir
Electronic publication date: 2025 Nov 25
Publication date: 2025
Volume: 13
Electronic Location ID: e20390
Received 2025 Feb 10; Accepted 2025 Oct 24
Copyright: ©2025 Huang et al.
Copyright year: 2025
Copyright holder: Huang et al.
License: This is an open access article distributed under the terms of the Creative Commons Attribution License, which permits unrestricted use, distribution, reproduction and adaptation in any medium and for any purpose provided that it is properly attributed. For attribution, the original author(s), title, publication source (PeerJ) and either DOI or URL of the article must be cited.
License URL: https://creativecommons.org/licenses/by/4.0/

Keywords: Hepatocellular carcinoma, Oral cancer overexpression 1, Prognosis, Competing endogenous RNA regulatory network, Immune cell infiltration, Immune checkpoints

Funding: National Key Research and Development Program of China 2023YFC3605100 Natural Science Foundation of Hubei Province 2023AFB685 National Natural Science Foundation of China 81873533 Key Research and Development Program of Hubei Province 2022BCA001 National Natural Science Foundation of China 82371599 This study was supported by the National Key Research and Development Program of China (2023YFC3605100), the Natural Science Foundation of Hubei Province (2023AFB685) and the National Natural Science Foundation of China (81873533) to Le Zhang; the Key Research and Development Program of Hubei Province (2022BCA001) and the National Natural Science Foundation of China (82371599) to Cuntai Zhang. The funders had no role in study design, data collection and analysis, decision to publish, or preparation of the manuscript.

==============================
Hepatocellular carcinoma (HCC) is a major cause of cancer-related mortality globally. Previous studies have reported that oral cancer overexpression 1 (ORAOV1) is overexpressed in HCC and correlated with poor prognosis, yet its molecular mechanisms remain incompletely understood. In this study, ORAOV1 overexpression was confirmed in HCC tissues via tissue microarray analysis and functionally linked to tumor cell proliferation through a positive correlation with Ki-67 expression in the human HCC cell line MHCC-97L. Bioinformatics analyses using The Cancer Genome Atlas (TCGA) and three Gene Expression Omnibus (GEO) HCC datasets further supported these findings. Multiple mechanisms appear to drive ORAOV1 upregulation, including promoter hypomethylation, amplification of the 11q13 region, and a putative ceRNA network involving AC005332.1, AC012615.1, and hsa-miR-100-5p. Gene Ontology and Kyoto Encyclopedia of Genes and Genomes (KEGG) pathway analyses implicated ORAOV1 in various cellular processes, such as abnormal membrane channel function, extracellular matrix–receptor interactions, IL-17 signaling, and peroxisome proliferator-activated receptor (PPAR) signaling. Co-expression analysis identified significant associations between ORAOV1 and the oncogenes TPCN2 and CCND1. Additionally, ORAOV1 expression correlated with enhanced infiltration of immunosuppressive cells, including regulatory T cells, myeloid-derived suppressor cells, and cancer-associated fibroblasts, as well as upregulation of immune checkpoint markers (PD-1, PD-L1, and CTLA-4). These results indicate that ORAOV1 may modulate the immunosuppressive tumor microenvironment and contribute to resistance against immunotherapy, highlighting its potential as a therapeutic target in HCC.

Introduction

Primary liver cancer, predominantly hepatocellular carcinoma (HCC), constitutes a significant global health challenge, ranking as the sixth most commonly diagnosed cancer and the fourth leading cause of cancer-related mortality worldwide (Shi et al., 2023; Sung et al., 2021). Accounting for 75%–85% of primary liver cancer cases, HCC is characterized by aggressive progression and unfavorable clinical outcomes (Gao et al., 2023). Chronic inflammation, driven by established risk factors including hepatitis B virus (HBV) infection, excessive alcohol consumption, and nonalcoholic fatty liver disease (NAFLD), represents a major etiological driver of HCC (Lawal et al., 2021). Although the contribution of NAFLD to HCC incidence is increasing, HBV infection remains the most prominent risk factor, accounting for approximately 50% of HCC cases globally (Akinyemiju et al., 2017).

Considerable progress has been made in the clinical management of HCC, with treatment modalities encompassing liver transplantation, surgical resection (DiNorcia et al., 2020; European Association for the Study of the Liver, 2018), percutaneous ablation (Llovet et al., 2021a), transhepatic arterial chemotherapy and embolization (TACE), radioembolization (Llovet & Bruix, 2003), and systemic therapy (Sangro et al., 2021). Treatment selection is guided by factors such as tumor burden, anatomical location, and underlying patient comorbidities (European Association for the Study of the Liver, 2018). Nevertheless, HCC incidence and mortality rates remain closely aligned worldwide, and the prognosis for patients, especially those with advanced disease, remains poor (Llovet et al., 2022).

Immunotherapy, particularly immune checkpoint blockade (ICB) targeting pathways such as PD-1/PD-L1 and CTLA-4, has emerged as a highly promising therapeutic approach for multiple cancers, including HCC (Rimassa, Finn & Sangro, 2023; Xing et al., 2021). ICB seeks to counteract tumor-mediated immunosuppression by reinvigorating the host’s antitumor immune response (Du et al., 2021). However, its efficacy in HCC is often constrained by primary or acquired resistance, observed in a substantial subset of patients (Kwong et al., 2025; Zhang et al., 2024). A major contributing factor is the immunosuppressive tumor microenvironment (TME), wherein aberrant expression and regulation of immune checkpoint molecules like PD-L1 play pivotal roles in treatment failure and adverse outcomes (Kong et al., 2023). The dynamic nature of PD-L1 expression further complicates the prediction of therapeutic response and resistance mechanisms (Cao et al., 2024). Thus, a deeper understanding of the molecular mechanisms underpinning immune evasion and ICB resistance in the HCC TME is essential for developing more effective treatment strategies.

The human genome and alterations within the TME are critically implicated in HCC pathogenesis (Llovet et al., 2021b; Rebouissou & Nault, 2020). Molecular and immune classification systems have been established to categorize HCC based on key driver mutations, signaling pathway activations, and immune contexture, integrating genomic, epigenomic, histopathological, and immunological data (Llovet et al., 2021b; Rebouissou & Nault, 2020). As an immunologically active organ, the liver contains diverse immune populations that exert context-dependent roles in HCC initiation, progression, treatment response, and prognosis, influenced by the composition and spatial distribution of immune infiltrates within the TME (Llovet et al., 2022; Ringelhan et al., 2018; Sangro et al., 2021).

Competing endogenous RNA (ceRNA) networks have recently emerged as important regulatory mechanisms in various malignancies, including liver cancer (Kong et al., 2019; Shi et al., 2021; Zhang et al., 2020). These networks involve long non-coding RNAs (lncRNAs) that act as molecular sponges for microRNAs (miRNAs), thereby modulating the expression of target genes (Bridges, Daulagala & Kourtidis, 2021; Salmena et al., 2011). Investigation of ceRNA interactions in HCC offers novel insights into potential therapeutic strategies, wherein targeting specific lncRNAs may enable more precise interventions.

Oral cancer overexpression 1 (ORAOV1) is located within the chromosome band 11q13, between cyclin D1 (CCND1) and fibroblast growth factor 19 (FGF19) (Huang et al., 2002). It has been reported to be overexpressed in multiple cancer types—including gastric, esophageal, and breast cancers—where it facilitates tumor growth and suppresses apoptosis (Ha et al., 2021; Jiang et al., 2010; Jiang et al., 2008; Kang & Koo, 2012; Komatsu et al., 2006; Turner et al., 2010; Zhai et al., 2014; Zucman-Rossi et al., 2015). In HCC, ORAOV1 knockdown has been shown to induce apoptosis, suppress proliferation, and inhibit tumor growth in both in vitro and in vivo models, suggesting a proto-oncogenic role (Ha et al., 2021; Jiang et al., 2010). ORAOV1 has also been proposed as a potential prognostic biomarker and therapeutic target in HCC (Ha et al., 2021). However, the mechanisms driving its overexpression and its precise functional contributions to HCC pathogenesis, particularly within the immune TME and in relation to immunotherapy resistance, remain poorly elucidated.

In this study, an in vitro HCC tissue microarray was employed, and data from public repositories, including The Cancer Genome Atlas (TCGA), Gene Expression Omnibus (GEO), and UALCAN, were integrated to validate ORAOV1 overexpression in HCC and its correlation with advanced disease stage. Through bioinformatic analyses, ORAOV1-associated genes and functional networks were identified, leading to the discovery of a novel ceRNA regulatory axis consisting of AC005332.1, AC012615.1, hsa-miR-100-5p, and ORAOV1, which was found to contribute to ORAOV1 upregulation. Furthermore, the relationship between ORAOV1 expression and the infiltration of pro-tumor immune cells, as well as the expression of immune checkpoint molecules, was investigated. The potential role of ORAOV1 in mediating immunotherapy resistance in HCC was also examined. The findings provide new insights into ORAOV1-related pathogenesis in HCC and suggest its potential utility as a diagnostic biomarker and therapeutic target.

Materials and methods

Further validation of ORAOV1 overexpression and its clinical relevance in HCC

To further validate the overexpression of ORAOV1 in HCC, mRNA, miRNA, and lncRNA microarray datasets from HCC patients were obtained from TCGA (Tomczak, Czerwińska & Wiznerowicz, 2015) using the TCGA biolinks package in R (version 4.2.1) (Colaprico et al., 2016). Additional HCC mRNA datasets (GSE45267: 62 tumors vs. 41 normals; GSE121248: 70 tumors vs. 37 normals) and miRNA datasets (GSE108724: seven paired tumors/normals; GSE69580: five paired tumors/normals) were downloaded from the Gene Expression Omnibus (GEO) (Barrett et al., 2013). The association between ORAOV1 expression and clinical features was evaluated using the UALCAN database (Zhang et al., 2022a). Statistical significance was defined as *p < 0.05; **p < 0.01; ***p < 0.001.

Clinical samples and tissue immunofluorescent

The HCC tissue microarray (catalogue No: HLiVH180Su17) was procured from Shanghai Outdo Biotech Co., Ltd. (Institutional Code: YB M-05-02; Shanghai, China) with approval from the Institutional Review Board (Reference No: SHYJS-CP-1710004). The microarray contained 108 tissue samples, comprising both adjacent non-cancerous tissues and paired cancerous tissues from 54 HCC cases. All participants were male, with a mean age of 47.20 ± 10.34 years. Immunofluorescence was performed on tissue sections using antibodies against Ki-67 (ab15580, 1:400) and ORAOV1 (CSB-PA003600, 1:200). Imaging was carried out using a fluorescence microscope following nuclear staining with 4′,6-diamidino-2-phenylindole (DAPI).

Cell culture and transfection

The human HCC cell line MHCC-97L was procured from Procell Life Science & Technology Co., Ltd. (Wuhan, China). Cells were maintained in high-glucose Dulbecco’s Modified Eagle Medium (DMEM; Invitrogen, Carlsbad, CA, USA), supplemented with 10% fetal bovine serum (FBS; Gibco, Waltham, MA, USA), at 37 °C in a humidified atmosphere containing 5% CO2.

To knock down the expression of ORAOV1, small interfering RNA (siRNA) was employed. The small interfering RNAs (siRNAs) targeting human ORAOV1 were designed as follows. The guide strand sequence was 5′-UGAACAUUGAGUAACGAACdTdT-3′, and the passenger strand sequence was 5′-GUUCGUUACUCAAUGUUCAdTdT-3′. The non-targeting scrambled sequences (Sense: UUCUCCGAACGUGUCACGU/dT//dT/; Antisense: ACGUGACACGUUCGGAGAA/dT//dT/) was used as a negative control (si-NC). All siRNA oligonucleotides were synthesized by Sangon Biotech (Shanghai, China). For transfection, cells in the exponential growth phase were seeded into 6-well plates at a density of 1 × 105 cells per well and allowed to adhere for 24 h. Prior to transfection, the culture medium was replaced with serum-free Opti-MEM (Gibco, Waltham, MA, USA) for an 8-hour starvation period. Transfection was then performed using Lipofectamine 3000 reagent (Invitrogen, Carlsbad, CA, USA) according to the manufacturer’s protocol, with a final siRNA concentration of 10 µM.

RNA extraction and quantitative real-time PCR

Total RNA was isolated from transfected cells using an RNA purification kit (Magen, China) following the manufacturer’s instructions. cDNA was synthesized from total RNA using the ThermoScript™ RT-PCR System (Invitrogen, Carlsbad, CA, USA). Quantitative real-time PCR (qRT-PCR) was subsequently performed to measure the mRNA expression levels of ORAOV1 and Ki-67. The primer sequences used are listed in Table S1. The reaction was carried out using a standard SYBR Green protocol on a real-time PCR system. GAPDH was used as an endogenous reference gene for normalization. The relative mRNA expression levels were calculated using the comparative 2−ΔΔCt method. Knockdown efficiency was confirmed by assessing ORAOV1 mRNA levels in siRNA-transfected cells relative to the si-NC group.

Construction of the lncRNA-miRNA-mRNA regulatory axis

Differentially expressed miRNAs (DEmiRNAs) and lncRNAs (DElncRNAs) were identified from TCGA data using the limma package in R (Ritchie et al., 2015). DEmiRNAs and DElncRNAs were selected based on an adjusted p value < 0.05 and |log2 fold change (FC)| > 0.5. Potential upstream miRNAs and lncRNAs interacting with ORAOV1 were predicted using the StarBase database (Li et al., 2014). Venn analysis was employed to identify key miRNAs. miRNA-mRNA, miRNA-lncRNA, and mRNA-lncRNA coexpression analyses were conducted using data from StarBase, with interactions retained based on a Pearson’s correlation |r| > 0.1 and p < 0.05. Based on the ceRNA network theory, a lncRNA-miRNA-mRNA regulatory axis was established (Pu et al., 2024).

Screening of differentially expressed mRNAs and gene enrichment analysis

Differentially expressed mRNAs (DEmRNAs) associated with ORAOV1 were identified using the limma package in R (Ritchie et al., 2015), with criteria of adjusted p < 0.05 and |log2FC| > 0.5. Gene Set Enrichment Analysis (GSEA) was conducted using the clusterProfiler package in R to explore the functional roles of ORAOV1-associated DEmRNAs, including Gene Ontology (GO) and Kyoto Encyclopedia of Genes and Genomes (KEGG) enrichment analyses (Subramanian et al., 2005; Yu et al., 2012). DEmRNAs with Pearson’s ratio (|r|) > 0.5 and p < 0.05 were considered significantly coexpressed with ORAOV1. Protein-protein interaction (PPI) networks of ORAOV1 and significantly coexpressed genes were constructed using the STRING database (https://www.string-db.org/) (Szklarczyk et al., 2019) , with a combined score >0.4 regarded as statistically significant.

Tumor immune analysis

The correlation between ORAOV1 expression and the infiltration of regulatory T cells (Tregs), myeloid-derived suppressor cells (MDSCs), and cancer-associated fibroblasts (CAFs) was analyzed using the TIMER2 database (Li et al., 2017). Additionally, the expression of immune checkpoints, including cytotoxic T lymphocyte-associated antigen 4 (CTLA4), programmed cell death protein 1 (PD1), and its ligand PD-L1, was examined for potential correlations with ORAOV1 (Tang et al., 2021), with a |r| > 0.1 and p < 0.05 considered statistically significant. The impact of ORAOV1 overexpression on the CTLA4 and/or PD1/PD-L1 immune checkpoint inhibitors (ICIs) was assessed using Immune Prediction Score (IPS) analysis on TCGA-HCC patients, with data obtained from the Cancer Immunohistology Atlas (TCIA) (Charoentong et al., 2017). The IPS is a predictive score used to estimate the likelihood of tumor response to ICIs, where a higher IPS indicates a better response to ICI treatment (Charoentong et al., 2017).

Statistical analysis

Immunofluorescence data were analyzed using SPSS version 22.0 (IBM Corp., Armonk, NY, USA). Paired t-tests were applied to compare ORAOV1 and Ki-67 expression between cancerous and adjacent non-cancerous tissues. Spearman’s rank correlation was used to assess the correlation between ORAOV1 and Ki-67 expression. A p-value < 0.05 was considered statistically significant.

Results

Potential association of ORAOV1 with HCC cell proliferation

The expression of ORAOV1 in HCC tissues and its potential correlation with cell proliferation were evaluated using a tissue microarray comprising paired cancerous and adjacent non-cancerous samples from 54 HCC cases. Immunostaining was performed with antibodies against ORAOV1 and Ki-67, a recognized marker of proliferative activity. ORAOV1 and Ki-67 expression were both significantly elevated in HCC tissues compared to adjacent normal tissues (Figs. 1A, 1B). A strong positive correlation between ORAOV1- and Ki-67-positive areas was observed (P < 0.001, R2 = 1) (Fig. 1C), suggesting a potential role for ORAOV1 in promoting tumor cell proliferation. To assess functional involvement, ORAOV1 was knocked down in HCC cells, resulting in a significant reduction in Ki-67 mRNA levels (Fig. 1D), indicating that ORAOV1 may regulate the expression of this proliferation-associated gene.

Figure 1 Potential association of ORAOV1 with HCC cell proliferation.

(A) Representative immunofluorescence images of ORAOV1 (Green) and Ki-67 (red) expression in HCC tissues (n = 54; scale bar, 50 µm)(B) The percentage of positive areas for ORAOV1 and Ki-67 in HCC tissues was analyzed using a paired t-test. (C) Spearman correlation analysis of Ki-67 and ORAOV1 expression levels in HCC tissues. (D) Relative mRNA expression of ORAOV1 and Ki-67 after transfection with control siRNA (si-NC) or ORAOV1-specific siRNA (si-ORAOV1) for 48 h, as determined by qRT-PCR. GAPDH was used for normalization. Data are mean ± SD of four replicates. ***p < 0.001.

ORAOV1 overexpression in HCC

ORAOV1 mRNA expression was analyzed across four independent HCC datasets, all of which showed significant upregulation in tumor tissues compared to normal controls (Figs. 2A–2D).

Figure 2 Overexpression of ORAOV1 in HCC.

The over expression of ORAOV1 mRNA in the H CC cohorts from the TCGA database (A), GSE56236 (B), GSE20140 (C) and GSE121248 datasets (D). ORAOV1 expression levels (transcript per million) stratified by tumor grade (E), nodal metastasis status (F), and ORAOV1 promoter methylation levels (G) in HCC. *p < 0.05; ***p < 0.001.

To further explore the clinical relevance of ORAOV1, its expression was correlated with clinicopathological features using the UALCAN database. Elevated ORAOV1 expression was associated with higher tumor grades and advanced metastatic status (Figs. 2E, 2F). Additionally, promoter methylation levels of ORAOV1 were significantly reduced in primary tumors compared to normal tissues (p < 0.001) (Fig. 2G), suggesting epigenetic involvement in its overexpression. These results indicate that ORAOV1 may serve as a prognostic biomarker in HCC.

Construction of a ceRNA Network involving ORAOV1 in HCC

Differentially expressed miRNAs (DEmiRNAs) in HCC were identified from the TCGA database, revealing 103 upregulated and 198 downregulated miRNAs (adj. p < 0.05, |log2FC| > 0.5; Fig. 3A, Table S2). Using the StarBase database, 72 miRNAs were predicted to target ORAOV1 (Table S3). Intersection with the downregulated miRNAs yielded 17 candidates potentially enhancing ORAOV1 expression (Fig. 3B, Table S4). Among these, only hsa-miR-29c-3p and hsa-miR-100-5p exhibited significant negative correlations with ORAOV1 (r < −0.1, p < 0.05; Table 1, Table S5). Prognostic analysis indicated that only hsa-miR-100-5p was associated with improved progression-free survival (PFS) and overall survival (OS) in HCC patients (Figs. 3C, 3D). Consistent with TCGA findings, independent GEO datasets (GSE45627, GSE121248 for ORAOV1; GSE108724, GSE69580 for miR-100-5p) confirmed dysregulation of these hub genes in HCC (Figs. 3E, 3F).

Figure 3 Construction of the ceRNA axis involving ORAOV1 in HCC.

(A) Volcano plot of differentially expressed miRNAs (DEmiRNAs) in HCC. Red and blue dots represent up- and down-regulated miRNAs, respectively. (B) Venn diagram identifying 17 overlapping miRNAs between the 198 downregulated DEmiRNAs and 72 ORAOV1-targeting miRNAs predicted by StarBase. (C, D) Overall survival and progression-free survival curves for hsa-miR-29c-3p (C) and hsa-miR-100-5p (D) in the TCGA-HCC cohort. (E) ORAOV1 mRNA expression in two independent GEO datasets (GSE45627, GSE121248), confirming significant upregulation in tumor tissues. (F) hsa-miR-100-5p expression in two additional GEO miRNA datasets (GSE108724, GSE69580), showing significant downregulation in HCC (*p < 0.05, **p < 0.01). (G) Volcano plot of differentially expressed lncRNAs (DElncRNAs) in HCC. Red and blue indicate up- and down-regulated lncRNAs, respectively. (H) Venn diagram showing two overlapping lncRNAs between the 495 upregulated DElncRNAs and 14 lncRNAs predicted to bind hsa-miR-100-5p. (I, J) Progression-free survival analysis for AC012615.1 (I) and AC005332.1 (J) in the TCGA-LIHC cohort. p < 0.05 was considered statistically significant. (K) Schematic representation of the proposed AC005332.1& AC012615.1/hsa-miR-100-5p/ORAOV1 ceRNA axis. Rectangle width reflects interaction strength.

Subsequently, differentially expressed long non-coding RNAs (DElncRNAs) in HCC were identified from the TCGA database, yielding 495 upregulated and 160 downregulated lncRNAs (Table S6). A volcano plot was generated using thresholds of |log2FC| > 1.5 and adj. p < 0.05 to visualize the DElncRNAs (Fig. 3G). Using the StarBase database, 14 upstream lncRNAs of hsa-miR-100-5p were predicted (Table S7). Intersection of the 495 upregulated DElncRNAs with these 14 candidates identified two lncRNAs, AC005332.1 and AC012615.1, both of which satisfied the co-expression criteria (|r| > 0.1, p < 0.05) within the ORAOV1-hsa-miR-100-5p ceRNA network (Fig. 3H, Table 1). Prognostic evaluation revealed that both lncRNAs were significantly associated with poor progression-free survival (PFS) in HCC (Figs. 3I, 3J). Based on these results, a ceRNA regulatory axis was proposed: AC005332.1 and AC012615.1/hsa-miR-100-5p/ORAOV1, which may contribute to ORAOV1 upregulation and unfavorable prognosis in HCC (Fig. 3K).

Identification of ORAOV1-related DEmRNAs and gene enrichment analysis

Differentially expressed mRNAs (DEmRNAs) associated with ORAOV1 were identified using the limma package in R (Ritchie et al., 2015). A total of 16,785 genes were upregulated (log2FC > 0.5, p < 0.05) and 323 were downregulated (log2FC < −0.5, p < 0.05) in tumors compared to normal tissues. Their distribution was visualized via a volcano plot (Fig. 4A), and a heatmap was generated to display the top 10 upregulated and downregulated genes (Fig. 4B, Table S8).

Figure 4 Enrichment analysis and functional networks of DEmRNAs related to ORAOV1 in HCC.

(A) Volcano plots describe 17,108 DEmRNAs with |log2 FC| > 0.5 and adj. p < 0.05. (B) Heatmap displaying the top 10 genes positively and negatively correlated with ORAOV1 in HCC. Red indicates upregulated genes, while blue represents downregulated genes. (C) Gene Ontology (GO) term enrichment for ORAOV1-related genes. (D) Kyoto Encyclopedia of Genes and Genomes (KEGG) pathway enrichment for ORAOV1-related genes. (E) Co-expression analysis of ORAOV1-related genes. (F) Protein–protein interaction (PPI) network of ORAOV1 and its co-expressed genes. Abbreviations: AFG3L1P, ATPase family gene-3, yeast-like-1; CCND1, cyclin D1; ECM, extracellular matrix; HCC, hepatocellular carcinoma; IGHMBP2, immunoglobulin mu-binding protein 2; IL-17, interleukin-17; ORAOV1, Oral cancer overexpression 1; PPAR, peroxisome proliferator activated receptor; TPCN2, two-pore channel 2.

Gene Set Enrichment Analysis (GSEA) (Subramanian et al., 2005) of the 17,108 DEmRNAs revealed significant enrichment in 37 KEGG pathways and 250 GO terms (p < 0.05, q < 0.25). Highly enriched GO terms included “ion channel complex”, “gated channel activity”, and “ion channel activity”, indicating involvement in membrane channel structure and function (Fig. 4C). Prominently enriched KEGG pathways included “ECM-receptor interaction”, “IL-17 signaling pathway”, and “PPAR signaling pathway” (Fig. 4D). The top 15 GO terms and KEGG pathways are provided in Table S9.

Table 1 Correlation analysis between mRNA and miRNA or lncRNA and miRNA or lncRNA and ORAOV1 in HCC determined by the starBase database.

miRNA	mRNA	Pearson’s r value	p value	
hsa-miR-29c-3p	ORAOV1	−0.122	0.0193*	
hsa-miR-100-5p	ORAOV1	−0.127	0.0147*	
lncRNA	miRNA	Pearson’s r value	p value	
AC005332.1	hsa-miR-100-5p	−0.105	4.40E−02*	
AC012615.1	hsa-miR-100-5p	−0.189	2.50E−04***	
lncRNA	mRNA	Pearson’s r value	p value	
AC005332.1	ORAOV1	0.317	3.33E−10***	
AC012615.1	ORAOV1	0.308	1.12E−09***	
Notes.

* p < 0.05.

*** p < 0.001

Co-expression analysis and PPI network construction

Co-expression analysis using the limma package identified four genes most strongly correlated with ORAOV1: cyclin D1 (CCND1), two-pore channel 2 (TPCN2), immunoglobulin mu-binding protein 2 (IGHMBP2), and ATPase family gene-3, yeast-like-1 (AFG3L1P) (Fig. 4E). A protein-protein interaction (PPI) network constructed via the STRING database suggested potential interactions between ORAOV1 and TPCN2 (score: 0.566) and CCND1 (score: 0.524) (Fig. 4F, Table S10).

Relationship between ORAOV1 and immune cell infiltration in HCC

The association between ORAOV1 expression and immune infiltration was assessed using TIMER2 (Li et al., 2017). ORAOV1 expression was significantly positively correlated with infiltration of regulatory T cells (Tregs) (Knochelmann et al., 2018; Noack & Miossec, 2014), myeloid-derived suppressor cells (MDSCs) (Gomez et al., 2020; Wesolowski, Markowitz & Carson 3rd, 2013), and cancer associated fibroblast cells (CAFs) (Affo, Yu & Schwabe, 2017) (p < 0.05) , but not with tumor purity (p = 0.0526) (Fig. 5A).

Figure 5 The correlation between ORAOV1 expression and immune cells in HCC.

(A) Correlation of ORAOV1 expression with tumor purity and the infiltrating levels of 3 tumor-promoting immune cells in HCC: Tregs, MDSCs and CAFs. (B) Correlations between ORAOV1 expression and 26 known immune checkpoints including PD1, PD-L1 and CTLA4 (p < 0.001 & r > 0.1). (C) Violin plots of the immunophenoscore (IPS) visualized the responses to CTLA4 and/or PD1/PD-L1 blocker treatment between the high and low ORAOV1 expression groups. None: immunotherapy without PD1/PD-L1 and CTLA4 blocker treatment. *p < 0.05; **p < 0.01; ***p < 0.001. CAFs, cancer-associated fibroblasts; CTLA4, cytotoxic T lymphocyte associated antigen 4; HCC, hepatocellular carcinoma; MDSCs, myeloid-derived suppressor cells; ORAOV1, oral cancer overexpression 1; PD1, programmed cell death protein 1; Tregs, regulatory T cells.

Human cancers, including HCC, evade antitumor immune responses by expressing the corresponding ligands of immune checkpoints in tumor and stromal cells (Topalian, 2017; Topalian et al., 2016). Using the GEPIA2 database (http://gepia2.cancer-pku.cn/#index) and Spearman correlation (|r| > 0.1, p < 0.05) (Tang et al., 2019), ORAOV1 expression was found to be positively correlated with 26 immune checkpoint genes, including CTLA4, PD1, and PD-L1 (Fig. 5B), all of which are commonly targeted in immunotherapies for HCC (Sangro et al., 2021; Topalian, 2017; Topalian et al., 2016) (Fig. 5B). These results imply a potential role for ORAOV1 in promoting an immunosuppressive microenvironment.

To further assess the potential impact of ORAOV1 expression on immunotherapy outcomes, immunophenoscore (IPS) analyses were performed (Charoentong et al., 2017). These analyses, which were stratified by different immunotherapy regimens involving CTLA4 and/or PD1/PD-L1 blockers, revealed that patients with high ORAOV1 expression had significantly lower IPS across regimens targeting CTLA4 and/or PD1/PD-L1 (p < 0.05), suggesting that ORAOV1 overexpression may be associated with reduced response to immune checkpoint inhibitors in HCC (Fig. 5C).

Discussion

In this study, overexpression of ORAOV1 in HCC tissues was initially validated using an HCC tissue microarray, and a causal correlation of ORAOV1 with tumor cell proliferation marker Ki-67 was observed in vitro in the a HCC cell line, indicating a potential association with tumor cell proliferation. Transcriptomic analyses across four independent HCC datasets confirmed significant upregulation of ORAOV1, consistent with previous reports of its overexpression in this malignancy (Ha et al., 2021). Furthermore, ORAOV1 expression was positively correlated with higher tumor grade and nodal metastasis status, supporting its potential role as a prognostic biomarker in HCC.

A reduction in DNA methylation at the ORAOV1 promoter was observed in HCC tissues compared to normal controls. This finding aligns with the well-established role of promoter hypomethylation in gene derepression (Lou et al., 2014), suggesting that aberrant upregulation of ORAOV1 in HCC may be partly attributable to epigenetic dysregulation in its promoter region.

To further investigate the regulatory mechanisms underlying ORAOV1 overexpression, a lncRNA-miRNA-mRNA ceRNA network was constructed, culminating in the proposed axis AC005332.1& AC012615.1/ hsa-miR-100-5p / ORAOV1. Using StarBase, hsa-miR-100-5p was identified as a putative upstream regulator of ORAOV1 and was found to be downregulated in HCC. Notably, low expression of hsa-miR-100-5p was associated with improved overall and progression-free survival, consistent with its previously documented tumor-suppressive roles in stomach adenocarcinoma (Wang et al., 2021), oral cancer (Henson et al., 2009), esophageal cancer (Zhang & Tang, 2017) and HCC (Shi et al., 2021; Song et al., 2019). Additionally, both AC012615.1 and AC005332.1 were significantly upregulated in HCC and correlated with poor progression-free survival. Subsequent validation confirmed that hsa-miR-100-5p serves as a key intermediary regulated by these lncRNAs. To our knowledge, only one study has suggested a protective role for AC012615.1 in glioblastoma (Yang et al., 2021), and no prior reports exist on AC005332.1. Thus, this study is the first to describe the AC005332.1& AC012615.1/hsa-miR-100-5p/ORAOV1 regulatory axis in HCC, providing novel insight into the molecular pathogenesis of this disease.

PPI analysis identified four genes strongly correlated with ORAOV1 in HCC samples: AFG3L1P, CCND1, IGHMBP2, and TPCN2. Among these, CCND1, IGHMBP2, and TPCN2 are co-located with ORAOV1 within the frequently amplified 11q13 chromosomal region (Grohmann et al., 2001; Huang et al., 2002; Khan et al., 2007), which has been implicated in HCC pathogenesis (Zhai et al., 2014; Zucman-Rossi et al., 2015). AFG3L1 is situated near the telomere on chromosome 16q24 (Shah et al., 1998). CCND1, a key cell cycle regulator, is commonly overexpressed or amplified in various cancers including HCC (Qie & Diehl, 2016), and its silencing has been shown to suppress liver cancer stem cell differentiation (Zhang, 2020). Amplification of CCND1 may also contribute to immunosuppression and poor response to immune checkpoint inhibitors in solid tumors (Chen et al., 2020). TPCN2, a Ca2+-permeable endolysosomal ion channel, suppresses HCC cell proliferation and tumor growth upon inhibition (Müller et al., 2021). To date, no direct association between IGHMBP2 and HCC has been reported.

A significant positive correlation was observed between ORAOV1 expression and levels of CTLA4, PD1, and PD-L1, as well as infiltration of tumor-promoting immune cells such as Tregs (Knochelmann et al., 2018; Noack & Miossec, 2014), MDSCs (Gomez et al., 2020; Wesolowski, Markowitz & Carson 3rd, 2013) and CAFs (Affo, Yu & Schwabe, 2017) in HCC. Immune checkpoint inhibitors targeting CTLA4 and PD-1/PD-L1 have become standard treatment for advanced liver cancer (Chae et al., 2018; Sangro et al., 2021). Tregs suppress antitumor immunity through interactions involving CTLA4 and PD-1/PD-L1 (Knochelmann et al., 2018; Noack & Miossec, 2014); MDSCs promote tumor progression and metastatic niche formation, and confer resistance to immunotherapy via suppression of T and NK cells (Gomez et al., 2020; Law, Valdes-Mora & Gallego-Ortega, 2020); and CAFs, central players in liver fibrosis within both pre-malignant and tumor microenvironments, drive HCC progression (Affo, Yu & Schwabe, 2017). IPS analysis indicated that high ORAOV1 expression was associated with significantly lower IPS under various anti-CTLA4 and/or anti-PD-1/PD-L1 regimens, suggesting that ORAOV1 may promote an immunosuppressive TME and contribute to primary resistance to immunotherapy. Thus, ORAOV1 expression may serve as a predictive biomarker for response to immune checkpoint blockade, and its targeted inhibition may represent a promising strategy to enhance immunotherapy efficacy in HCC.

While previous work by Ha et al. (2021) demonstrated that silencing ORAOV1 suppresses HCC migration, invasion, and xenograft growth, the mechanistic basis remained unclear. In this study, GO enrichment analysis indicated that ORAOV1-associated genes are involved in membrane channel structure and function. KEGG analysis further revealed enrichment in ECM-receptor interaction, IL-17 signaling, and PPAR signaling pathways.

The extracellular matrix (ECM) is a critical modulator of the TME and influences immunotherapy response in HCC (Mohan, Das & Sagi, 2020; Ringelhan et al., 2018). IL-17, a proinflammatory cytokine, plays context-dependent roles in cancer—its dysregulation promotes immunopathology, autoimmunity, and tumor progression (Amatya, Garg & Gaffen, 2017). In mouse models, inhibition of IL-17 signaling reduced alcohol-induced HCC progression by suppressing PPARγ/PGC1-dependent cholesterol synthesis (Ma et al., 2020; Zhang et al., 2022b). IL-17 signaling has also been linked to resistance to immune checkpoint inhibitors (Chen et al., 2022). Recent evidence suggests that IL-17 induces collagen deposition in the ECM, shielding tumor cells from immune attack and conferring resistance to anti-PD-1/PD-L1 therapy in squamous cell carcinoma (Chen et al., 2022).

Moreover, increased ECM stiffness disrupts ion channel function and signal transduction, promoting tumor progression, immune evasion, and therapy resistance (Jiang et al., 2022). Based on these findings, we propose that ORAOV1-associated activation of IL-17 signaling may drive ECM remodeling, TME dysfunction, impaired channel activity, and broad therapeutic resistance in HCC (Fig. 6). Thus, combining conventional immunotherapy with anti-ORAOV1 or anti-IL-17 agents may represent a more effective therapeutic strategy.

Figure 6 Schematic representation of the mechanisms underlying the upregulation and pro-tumor functions of ORAOV1 in hepatocellular carcinoma.

CAFs, cancer-associated fibroblasts; CCND1, cyclin D1; CTLA4, cytotoxic T lymphocyte associated antigen 4; ECM, extracellular matrix; IGHMBP2, immunoglobulin mu-binding protein 2; IL-17, interleukin-17; MDSCs, myeloid-derived suppressor cells; ORAOV1, oral cancer overexpression 1; PD1/PD-L1, programmed cell death protein 1 and its ligand 1; PPAR, peroxisome proliferator activated receptor; TPCN2, two-pore channel 2; Tregs, regulatory T cells.

Beyond confirming ORAOV1 overexpression, this study provides a multidimensional framework with clinical relevance. The strong correlation between ORAOV1 and aggressive tumor features, immunosuppressive TME, and poor response to anti-CTLA4/PD-1 therapy positions ORAOV1 as a dual-function biomarker suitable for prognostic stratification and treatment response prediction. The newly identified ceRNA axis (AC005332.1 and AC012615.1/hsa-miR-100-5p/ORAOV1) offers novel therapeutic opportunities, such as miRNA mimics or lncRNA inhibitors, to suppress ORAOV1 and inhibit HCC progression.

Moreover, our results propose a unifying hypothesis connecting ORAOV1 to both proliferation and immunotherapy resistance via IL-17/ECM-mediated mechanisms. This model provides a strong rationale for evaluating combination therapies targeting ORAOV1 or IL-17 to sensitize resistant HCC to immune checkpoint blockade. Future preclinical and clinical studies are urgently needed to evaluate whether anti-ORAOV1 or anti-IL-17 agents can enhance the efficacy of existing immunotherapies and improve outcomes in HCC patients.

In conclusion, our study establishes ORAOV1 as a significantly overexpressed oncogene in HCC, validated through integrated in vitro experimental studies and bioinformatics approaches. ORAOV1 upregulation occurs via multiple mechanisms, including the AC005332.1 and AC012615.1/hsa-miR-100-5p/ORAOV1 ceRNA regulatory axis. Elevated ORAOV1 expression correlates strongly with aggressive clinicopathological features and promotes an immunosuppressive tumor microenvironment characterized by increased infiltration of pro-tumor immune cells and elevated expression of immune checkpoints such as CTLA4, PD1, and PD-L1. Furthermore, we identify IL-17-mediated ECM remodeling and TME stiffening as key downstream effects contributing to immunotherapy resistance and disease progression. Beyond its role as a dual prognostic and predictive biomarker, ORAOV1 represents a promising therapeutic target. Our findings provide a rationale for novel combination treatments targeting ORAOV1 or its associated pathways to overcome resistance to current immunotherapies in HCC.

Supplemental Information

Supplemental Information 1 Raw data

Supplemental Information 2 Supplemental tables

Additional Information and Declarations

Competing Interests

Author Contributions

Human Ethics

Data Availability

The authors declare there are no competing interests.

Yuzhen Huang conceived and designed the experiments, performed the experiments, analyzed the data, prepared figures and/or tables, authored or reviewed drafts of the article, and approved the final draft.

Ni Yang performed the experiments, analyzed the data, prepared figures and/or tables, authored or reviewed drafts of the article, and approved the final draft.

Su Wen performed the experiments, analyzed the data, prepared figures and/or tables, authored or reviewed drafts of the article, and approved the final draft.

Ziwei Fang performed the experiments, analyzed the data, prepared figures and/or tables, and approved the final draft.

Yucong Zhang performed the experiments, analyzed the data, prepared figures and/or tables, and approved the final draft.

Zonghao Qian performed the experiments, analyzed the data, prepared figures and/or tables, and approved the final draft.

Yi Huang performed the experiments, analyzed the data, prepared figures and/or tables, and approved the final draft.

Tiejun Yin conceived and designed the experiments, authored or reviewed drafts of the article, and approved the final draft.

Cuntai Zhang conceived and designed the experiments, authored or reviewed drafts of the article, and approved the final draft.

Le Zhang conceived and designed the experiments, analyzed the data, prepared figures and/or tables, authored or reviewed drafts of the article, and approved the final draft.

The following information was supplied relating to ethical approvals (i.e., approving body and any reference numbers):

Ethics Committee of Shanghai Outdo Biotech Company (Institutional Code: YB M-05-02) with approval from the Institutional Review Board (Reference No: SHYJS-CP-1710004).

The following information was supplied regarding data availability:

The raw data is available in the Supplemental Files.

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
