# Peer review of "Overexpression of ORAOV1 and its association with immunotherapy resistance in hepatocellular carcinoma"

_PeerJ, doi:10.7717/peerj.20390_

## Round 0.1 · original submission · Major Revisions

· Academic Editor

Major Revisions

**Language Note:** The review process has identified that the English language must be improved. PeerJ can provide language editing services - please contact us at [email protected] for pricing (be sure to provide your manuscript number and title). Alternatively, you should make your own arrangements to improve the language quality and provide details in your response letter. – PeerJ Staff

Reviewer 1 ·

Basic reporting

none

Experimental design

none

Validity of the findings

none

Additional comments

This study provides evidence that ORAOV1 is overexpressed in HCC, as demonstrated through both in vitro tissue microarray analysis and bioinformatics assessments. ORAOV1 upregulation occurs via multiple mechanisms, including the AC005332.1&AC012615.1/hsa-miR-100-5p/ORAOV1 ceRNA regulatory axis. Elevated ORAOV1 expression promotes the infiltration of tumor-associated immune cells, fosters resistance to immunotherapy, and correlates with poor prognosis in HCC. These effects are mediated by the upregulation of immune checkpoint molecules (e.g., CTLA4, PD1/PD-L1), as well as IL-17-driven alterations in the extracellular matrix (ECM) and tumor microenvironment (TME), leading to ECM/TME stiffening and related pathophysiological changes. Taken together, ORAOV1 may serve as a valuable prognostic biomarker for HCC and a potential therapeutic target to enhance the effectiveness of immunotherapy in this malignancy. It is interesting. However, the following issues need to be addressed,
1. In the preface part, the author needs to make a detailed on the current research progress of liver cancer prognosis summary, put forward the research purpose and meaning, when necessary, can consider to refer to the following documents: DOI 10.3389/fendo.2023.1153802. DOI 10.3389/fendo.2023.1153802

2. The expression and function of hub genes should be further verified through multiple public data sets
3. The discussion section needs to elaborate on the value of the research in light of the findings
4. The description of the method should be as concise as possible, and some key models need to cite some references
5. The presentation of charts should be further typeset and the resolution should be increased. Most charts cannot obtain information quickly
6. The language needs to be polished by professionals

·

Basic reporting

The authors used bioinformatic methods to reveal that the expression, mechanism, and function of ORAOV1 in HCC. While there has been a report of ORAOV1’s role in HCC (33161447), this paper produced something new about it. I have no doubt about the methods used in the study, but I still have 2 questions that may be important for the decision.
1. The ethnic approval was obtained from a commercial company? Where did this company collect the patients’ tissues? Is it authorized? Do the patients know and agree with this?
2. The authors showed the most powerful relationship between ORAOV1 and Ki-67 (R2=1), but the causality is still unknown. Could the authors simply validate the relationship in HCC cells? Also, the data-explored mechanism should be validated, which will make the conclusion more convinced.

Experimental design

none

Validity of the findings

none

---

## Round 0.2 · accepted · Accept

· Academic Editor

Accept

All concerns of the reviewers were addressed, and the revised manuscript is acceptable now.

·

Basic reporting

-

Experimental design

-

Validity of the findings

-

Additional comments

I have no other concern.